# Preparation of Polyvinyl Alcohol/Bacterial-Cellulose-Coated Biochar–Nanosilver Antibacterial Composite Membranes

**Liang Zhang [1,2,***, Sen Zheng [1], Zhihui Hu [1], Lvling Zhong [1], Yao Wang [1], Xiaomin Zhang [3] and Juanqin Xue [1]**

1   School of Chemistry and Chemical Engineering, Xi'an University of Architecture and Technology, Xi'an 710055, Shaanxi, China; pis20323c@126.com (S.Z.); huzhihui@live.xauat.edu.cn (Z.H.); idopretty2004@yahoo.co.jp (L.Z.); wyspacestar@aliyun.com (Y.W.); huagong1985@163.com (J.X.)
2   Shanxi Provincial Key Laboratory of Gold and Resource, Xi'an University of Architecture and Technology, Xi'an 710055, Shaanxi, China
3   School of Resources Engineering, Xi'an University of Architecture and Technology, Xi'an 710055, Shaanxi, China; xmzhang@xauat.edu.cn
*   Correspondence: zhangliang@xauat.edu.cn; Tel.: +86-1399-196-0740



**Featured Application:** **(1) Composite membrane with both filtration and antibacterial function. (2) PVA mechanical performance is stronger by adding BC. (3) Reuse of resources by using waste corn stover as biochar material. (4) C-Ag structure is more stable, Ag$^+$ release ability is stronger.**

**Abstract:** Pathogenic bacteria and microorganisms in drinking water can cause various diseases, and new types of antibacterial material for water treatment and filtration are urgently needed. In this work, polyvinyl alcohol/bacterial cellulose/biochar–nanosilver (PVA/BC/C-Ag) antibacterial composite membrane materials were prepared by uniformly dispersing C-Ag particles in a PVA/BC mixed gel. Fourier-transform infrared spectroscopy (FT-IR), X-ray diffractometry (XRD), scanning electron microscopy (SEM), and thermogravimetric differential scanning calorimetry (TG-DSC) were used to characterize the composite membranes. Results indicated that the BC was uniformly mixed into the PVA gel and that the C-Ag particles were uniformly immobilized in the PVA/BC hybrid membrane. The PVA/BC/C-Ag composite membranes exhibited excellent antibacterial activity against *Escherichia coli* when assayed using a plate-counting technique. When used to treat actual contaminated water, the composite membranes demonstrated sustained antibacterial activity and good reusability. PVA/BC/C-Ag composite membranes have great potential for the development of drinking water treatment applications.

**Keywords:** antibacterial activity; polyvinyl alcohol; bacterial cellulose; drinking water; silver nanoparticles

## 1. Introduction

Drinking water safety is an increasing global concern, and fresh water availability is particularly important in areas that lack surface water. In many developing countries, residents' drinking water is stored in reservoirs such as high-rise water tanks that can easily breed bacteria and other microorganisms that cause intestinal infections [1]. This presents a threat to the health and safety of residents [2] if not handled in a timely manner.

Filter materials commonly used in current drinking water terminal devices include activated carbon, resin, and filter membranes [3]. Terminal water treatment materials fall into two broad

categories: adsorbent materials [4] and membrane filtration materials [5]. However, such materials have no effect on the growth of microorganisms during water storage. Indeed, they eventually lose their water purification effects because they themselves become a breeding ground for microbial growth. An antibacterial complex composed of silver and carbon (Ag/C) was recently shown to provide good bactericidal action against *Bacillus subtilis* and *Escherichia coli* [6]. However, the antibacterial effect was not obvious, due to the weak bond between carbon and silver. When the content of $Fe_3O_4$ and straw is increased, a straw/$Fe_3O_4$/polycaprolactone complex has also shown strong antibacterial ability [7], but it cannot be directly applied to a terminal drinking water device.

Membrane filtration mainly involves the interception of larger particles and adsorbents in terminal water, but the membrane itself does not kill microorganisms. Song et al. [8] reported a multifunctional system for high-efficiency water treatment that combines traditional pressure-driven membrane filtration with solar thermal technology based on a photothermal membrane. Chen et al. [5] described a mesoporous, three-dimensional (3D) wood membrane decorated with palladium nanoparticles for efficient water treatment. Organic, biocompatible membranes were used to remove heavy metal ions and dyes such as lead and cadmium by Derami et al. [9]. However, most membrane materials only remove micro-contaminants and intercept large particles [5]. There are often bacterial contaminants in drinking water [1], and the eradication of these harmful bacteria has been neglected. Although Zhang et al. [10] reported that a chitosan/polyvinyl alcohol/$TiO_2$/Ag coating for a drinking water tank had an antibacterial effect, it had no filtering effect. To permit water to be drunk directly from the terminal, it is necessary to intercept and kill bacterial microorganisms. Development of a bacteriostatic filtration membrane that is highly efficient, long-lasting, and harmless to human health is therefore an urgent necessity.

Here, we describe a polyvinyl alcohol/bacterial cellulose/nanosilver-loaded biochar (PVA/BC/C-Ag) composite membrane with good antibacterial properties that can be synthesized using a simple method. PVA is used in many industries because the crosslinked polyvinyl alcohol gel has good biocompatibility, excellent mechanical properties [11,12], and a 3D network structure. However, by itself, PVA is prone to swelling and rupture when it encounters water. Bacterial cellulose has a fine 3D network structure, high tensile strength, biodegradability, and biocompatibility [13,14]; it is used in many fields, including the healthcare and cosmetics industries [15]. BC is also widely used because it enhances the mechanical strength of other materials when it is added to a composite [16]. Mixing PVA and BC can therefore improve the mechanical performance of the PVA membrane. Silver is widely used in antibacterial composite membranes because of its broad-spectrum bactericidal action and durability. However, excess silver ions in drinking water can also endanger human health. When corn stalk biochar is combined with silver nanoparticles, the resulting material has good coating properties and silver ion release characteristics [17,18].

In the present work, we used silver as a bacteriostatic additive to prepare silver-loaded biochar (C-Ag) by impregnation and high-temperature carbonization reduction. We predicted that the combination of carbon and silver would produce strong, long-lasting antibacterial properties and that the addition of carbon would affect the performance of a composite membrane. We synthesized an antibacterial PVA/BC/C-Ag composite membrane with a fine 3D network structure and characterized its mechanical properties by Fourier-transform infrared spectroscopy (FT-IR), X-ray diffractometry (XRD), scanning electron microscopy (SEM), and thermogravimetric differential scanning calorimetry (TG-DSC). We documented its antibacterial properties, assessed its performance in a simulated contaminated water treatment, and described potential mechanisms of its antibacterial action.

## 2. Experimental Section

### 2.1. Materials

Chemicals and materials: All reagents were of analytical grade. Nutrient agar, beef extract, and peptone were purchased from Beijing Aobo Biotechnology Co., Ltd, Beijing, China. Silver nitrate was

purchased from Shanghai Shenbo Chemical Co., Ltd, Shanghai, China. Polyvinyl alcohol (PVA-124, M = $(44.05)_n$) was purchased from Guangzhou Jinhua Chemical Reagent Co., Ltd, Guangzhou, China. BC emulsion was obtained by cultivating BC membranes and breaking them apart as described below. The C-Ag material was prepared by carbonizing corn stover soaked in $AgNO_3$ solution in a high temperature tube furnace. Other common chemical reagents were purchased from Tianjin Komiou Chemical Reagent Co., Ltd, Tianjin, China. *E. coli* were purchased from Xi'an Institute of Microbiology and used as a model microorganism for antibacterial activity trials.

### 2.2. Preparation of Composite Membranes

#### 2.2.1. Preparation of Bacterial Cellulose (BC) and BC Emulsion

*Gluconacetobacter xylinus* were cultivated in Hestrin–Schramm (HS) medium for BC production. The HS medium consisted of 25 g/L glucose, 7.5 g/L yeast extract, 10 g/L peptone, and 10 g/L disodium hydrogen phosphate in a 500 mL Erlenmeyer flask. The pH of the medium was adjusted to 4.0–5.0 by addition of acetic acid. The medium was then autoclaved at 125 °C for 20 min. The sterile medium was inoculated with *G. xylinus* from stock culture and incubated at 30 °C for 72 h under static conditions.

After 7 days of incubation, a white gelatinous pellicle (bacterial cellulose) had developed at the air–liquid interface of the Erlenmeyer flask. The pellicle was isolated and washed with deionized water, then mixed with 1% NaOH and boiled for 2 h at 80 °C to remove adherent growth medium and cells. The resultant film was rendered neutral by repeated washes with sterile distilled water. Finally, the purified cellulose was dried at room temperature and stored for subsequent experiments. The BC film was dispersed into a suspension using a tissue mashing homogenizer, and the suspension was further ultrasonicated for 10 min to obtain a BC emulsion.

#### 2.2.2. Preparation of the C-Ag Composites

5.0 g of corn stalks were immersed in 200 mL of $AgNO_3$ aqueous solution (0.1M), washed and dried after 24 h, and placed in a ceramic boat. It was then placed in a nitrogen-filled tube furnace and heated to 900 °C at a rate of 10 °C/min for 1 h. Finally, the sample was cooled, ground, and sieved through a 60 mesh, and the sample was marked as C-Ag900 [19].

#### 2.2.3. Preparation of PVA/BC/C-Ag Composite Membranes

PVA solution was prepared by dissolving 5 g PVA in 50 mL of deionized water. Next, 10 g of BC emulsion was added, and the solution was stirred for 30 min until it became homogeneous. Any bubbles were removed by ultrasound. Next, 0.1 g of C-Ag900 was added and the mixture was stirred for 60 min. Following deaeration, the resulting casting solution was coated onto glass at room temperature (dimensions $60 \times 60 \times 1$ mm, length × width × thickness) and placed in a hot blast-drying oven at 50 °C. After 4 h, the material had formed a membrane of approximately 0.2 mm thickness and was removed.

### 2.3. Characterization of PVA/BC/C-Ag Composite Membranes

A Nicolet iS50 ATR infrared spectrometer (Thermo Scientific Co., Shanghai, China) was used to characterize the Fourier-transform infrared spectra (FT-IR) of PVA, PVA/BC, and PVA/BC/C-Ag particulate materials in the 500 $cm^{-1}$ to 4000 $cm^{-1}$ wavenumber range. The crystal structure, bonding state, and calculated particle size of the composites were analyzed by X-ray diffractometry (XRD; Japanese Rigaku Ultimate Type IV, Beijing, China) with a scanning angle range from 0° to 90°. The surface morphology of the composites was observed by scanning electron microscopy (SEM; FEI Verios 460). The Brunauer–Emmett–Teller (BET) specific surface area of the material was determined by nitrogen adsorption–desorption isotherm measurements at 77 K (ASAP 2460). The stability of the PVA/BC/C-Ag composite was investigated by thermogravimetric differential scanning calorimetry (TG-DSC; LABSYS Evo, Setaram, France). Zeta potentials of samples were measured by Malvern

Zetasizer Nano ZS90 analyzer (Malvern Instruments Ltd. Shanghai, China) at pH 6.8 (dispersed in deionized water).

### 2.4. Mechanical Properties of PVA/BC/C-Ag Composite Membranes

For these experiments, composite membranes were cut to a size of 60 mm × 40 mm. Tensile tests of PVA/C-Ag and PVA/BC/C-Ag composite membranes were performed at room temperature using a WDW-BO5 universal testing machine at a stretching speed of 5 mm min$^{-1}$.

### 2.5. Swelling Performance of PVA/BC/C-Ag Composite Membranes

A swelling ratio test was carried out for the PVA membranes and for multiple formulations of PVA/(BC)$_x$/(C-Ag)$_y$ membranes (x = 0 g, 5 g, 10 g, 15 g; y = 0 g, 0.05 g, 0.10 g, 0.15 g, 0.20 g, 0.25 g). After weighing each dry membrane ($M_0$), it was immersed in deionized water, allowed to fully swell, and weighed again after swelling ($M$). Swelling ratio was calculated as:

$$\text{Swelling ratio} = \frac{M_0 - M}{M_0} \times 100\% \tag{1}$$

where $M_0$ is the initial dry membrane mass and $M$ is the membrane mass after swelling.

### 2.6. Antibacterial Activity of PVA/BC/C-Ag and PVA/BC Composite Membranes

PVA/BC/C-Ag and PVA/BC composite membranes with different contents of C-Ag and BC were cut into 9 mm diameter disks and placed on a solid plate medium coated with an *E. coli* bacterial suspension. The plates were cultured at 37 °C in a constant temperature biochemical incubator. After 24 h, the size of the antibacterial zone was measured.

### 2.7. Silver Loss from PVA/BC/C-Ag Composite Membranes

The national drinking water standard (GB5749-2006, China) for release of Ag$^+$ is <0.05 mg/L. Therefore, it was important to test the amount of silver released into water from the composite membranes. Group A membranes were immersed in water for 30 days, and the concentration of silver ions in the water was measured every five days using a graphite furnace atomic absorption spectrometer (GFAAS). Group B membranes were immersed in water for 10 time (each time is 6 h), and the silver ion concentrations were successively measured in the water.

### 2.8. Antibacterial Persistence of PVA/BC/C-Ag Composite Membranes

The persistence of the composite membranes' antibacterial properties and the reusability of the membranes themselves are important factors for their eventual use in water treatment. To assess antibacterial persistence, PVA/BC/C-Ag composite membranes were immersed in a bacterial liquid ($2.1 \times 10^5$ CFU/mL *E. coli*) for 3, 7, 10, 15, 20, 25, or 30 days under continuous monitoring. A 0.2 mL bacterial suspension was pipetted onto the solid medium at intervals. After 24 h, the number of colonies was calculated using the coating plate counting method and the formula was used to calculate its antibacterial properties. To test reusability, composite membranes were sterilized and immersed for 24 h in the same concentration of bacterial suspension. After that, they were washed and activated with dilute nitric acid (0.5 M), then rinsed with alcohol. This immersion was repeated five times. The above operation was carried out by taking 0.2 mL of the bacterial suspension each time. In both cases, the antibacterial ratio was calculated as

$$\text{Antibacterial ratio} = \frac{A_0 - A}{A_0} \times 100\% \tag{2}$$

where $A_0$ is the initial number of colonies and $A$ is the number of colonies after bacteriostasis.

*2.9. Antibacterial Activity of PVA/BC/C-Ag Composite Membranes in Actual Water*

To simulate real-world conditions, the PVA/BC/C-Ag composite membrane was used to filter actual contaminated water containing $2.3 \times 10^5$ CFU mL$^{-1}$ of *E. coli*. The membrane was cut into a 50 mm diameter circle and placed into a sterilized suction filtration device. The contaminated water was treated by filtration. After filtering specific water volumes, 0.1 mL of the filtered bacterial suspension was sampled and applied to a sterilized plate medium. After 24 h of incubation, the antibacterial rate was calculated by the plate-counting method as described above. The antibacterial activity of several membranes was tested using this method.

## 3. Results and Discussion

*3.1. Characterization of PVA/BC/C-Ag Composite Membranes*

3.1.1. Fourier-Transform Infrared (FT-IR) Spectrometry and X-ray Diffraction Analysis

FT-IR spectra were used to characterize PVA, PVA/BC, and PVA/BC/C-Ag membranes. The shapes and positions of most peaks in Figure 1a are similar. The pure PVA showed the characteristic PVA peak at 3268 cm$^{-1}$ due to the stretching vibration of O–H [20]. Peaks at 1415 cm$^{-1}$ and 1085 cm$^{-1}$ corresponded to the bending vibrations of C–O–C and C–H in PVA [21]. Comparing the curves of pure PVA and PVA-BC, the stretching vibration peak of the hydroxyl group shifted from 3268 cm$^{-1}$ to 3261 cm$^{-1}$, probably due to hydrogen bond vibration between the PVA and BC molecules. Comparing the curves of PVA-BC and PVA/BC/C-Ag, the stretching vibration peak of the hydroxyl group shifted from 3261 cm$^{-1}$ to 3265 cm$^{-1}$, which may be ascribed to the coordination between -OH and Ag. After the introduction of C-Ag, no new peaks appeared in the PVA/BC/C-Ag spectrum compared to the PVA/BC spectrum.

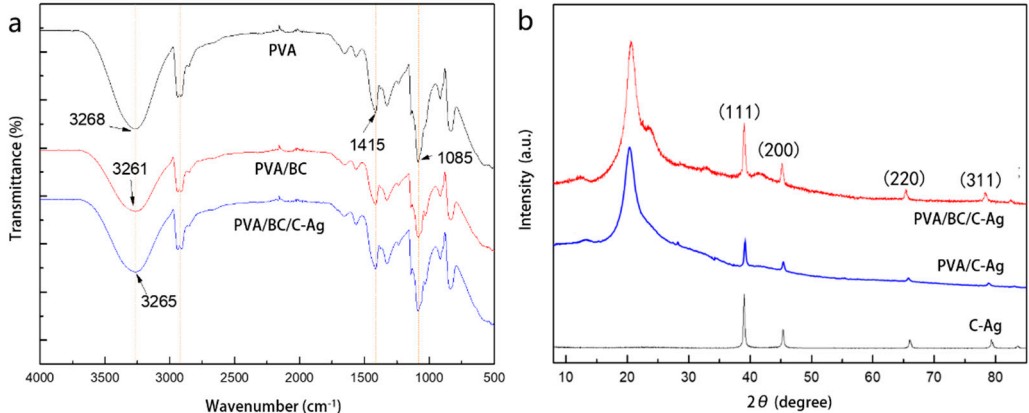

**Figure 1.** Fourier infrared analysis of several membranes (**a**), and X-ray diffraction analysis of different materials (**b**).

As shown in Figure 1b, the diffraction peaks of silver appeared at the same positions in the PVA/BC/C-Ag, PVA/C-Ag, and C-Ag composite membranes. The positions of the diffraction peaks at 2θ were 38.18° (111), 44.34° (200), 64.46° (220), and 77.47° (311), consistent with the standard pattern of Ag (JCPDS No 04-0783). Calculated using the Xie Le formula, the particle size of silver was approximately 50 nm. PVA had a strong diffraction peak at 2θ = 20° [22] and exhibited a weak characteristic peak at 2θ = 41°. The reason for the occurrence of diffraction was mainly the influence of the microcrystalline peak of the PVA molecule on the (101) crystal plane. The PVA/BC/C-Ag and PVA/C-Ag composite membranes differed in the occurrence of a weak characteristic peak at 2θ = 22°, which may have been a characteristic peak of PVA/BC. A weak characteristic peak near 2θ = 12° may have been a characteristic peak of BC [23].

### 3.1.2. SEM Analysis

Pure PVA, PVA/BC, C-Ag, PVA/C-Ag, and PVA/BC/C-Ag were characterized by SEM (Figure 2). The PVA/BC membrane (Figure 2b) had a looser surface topography than the pure PVA membrane (Figure 2a) because the three-dimensional structure of the PVA was split by filling with the BC emulsion. The BC emulsion appeared to be well-combined with the PVA. The silver nanoparticles exhibited a spherical shape and were well-embedded in the surface of the biochar (Figure 2d). The surface of the PVA/C-Ag membrane had spherical protrusions without scaly material because the biochar of the C-Ag material was coated in the PVA membrane, leaving some bare silver nanoparticle protrusions (Figure 2c). Comparing Figure 2c (PVA/C-Ag) and 2e (PVA/BC/C-Ag), it was confirmed that parts of the silver nanoparticles were exposed outside of membrane surface and that the structure of the PVA/C-Ag membrane was more dense.

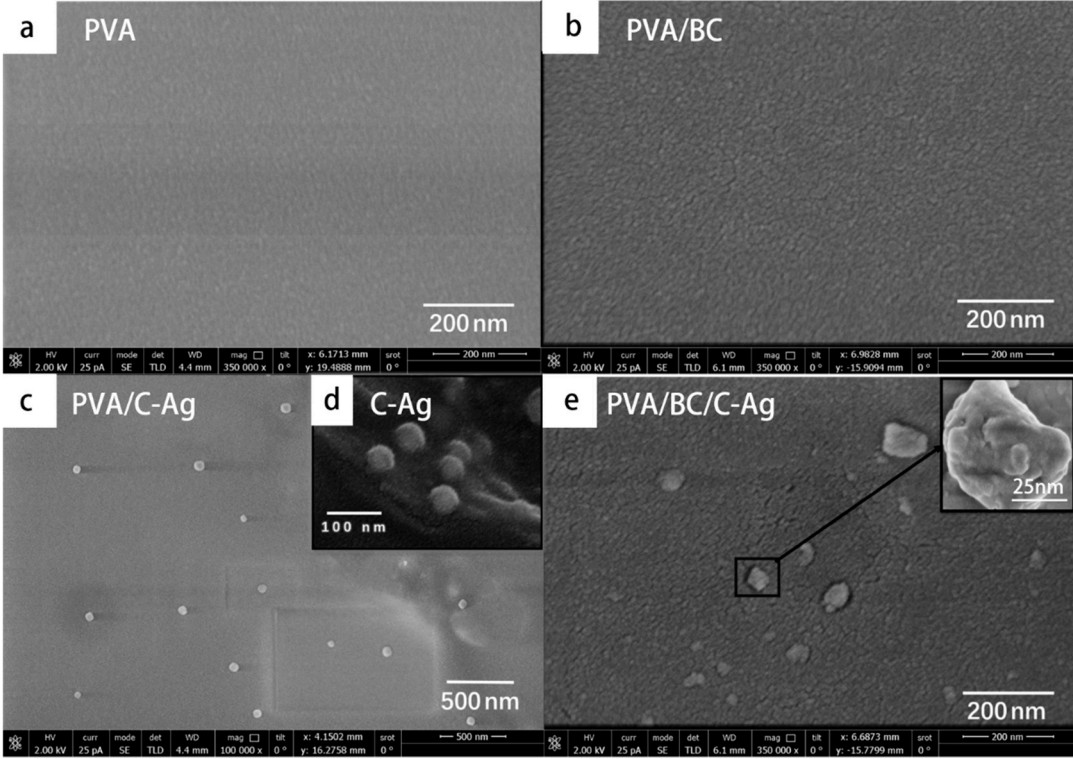

**Figure 2.** SEM images PVA (**a**), PVA/BC (**b**), PVA/C-Ag (**c**), C-Ag (**d**), and PVA/BC/C-Ag (**e**).

The results of the BET measurement (Supplementary Table S1) showed that due to the loading of silver on the surface of biochar and the encapsulation of PVA/BC, although the average pore sizes of C, C-Ag, and PVA/BC/C-Ag are increased, the surface area and the total pore volume appeared to decrease.

### 3.1.3. TG and DSC Analysis

Figure 3 shows the thermogravimetric and differential thermal curves of a PVA/BC/C-Ag composite membrane. Figure 3a shows that the membrane's weight loss could be divided into three stages. The first stage occurred between 100 and 150 °C and was mainly caused by the evaporation of water from the material. The endothermic peak appeared, and the weight loss is approximately 5%. The second stage occurred between 200 and 300 °C; the weight loss was approximately 60%, mainly due to the decomposition of PVA/BC polymer side chains [24]. The DSC curve shown in Figure 3b shows that the PVA/BC/C-Ag composite membrane had endothermic peaks near 200 °C and 250 °C, which may have been caused by the fusion of the BC crystal structure and the melt-phase transition of PVA.

The third stage occurs between 350 and 500 °C, mainly due to the fracture of C–C chains in the PVA/BC polymer membrane [25] and so-called carbonization; total weight loss was about 88%. The final weight remained stable, with the remaining components being mainly C-Ag and a small amount of charred PVA/BC ash. The thermogravimetric and differential thermal curves of the PVA/BC/C-Ag composite membrane indicated that the composite membrane will have excellent stability in actual water treatment where temperatures do not exceed 100 °C.

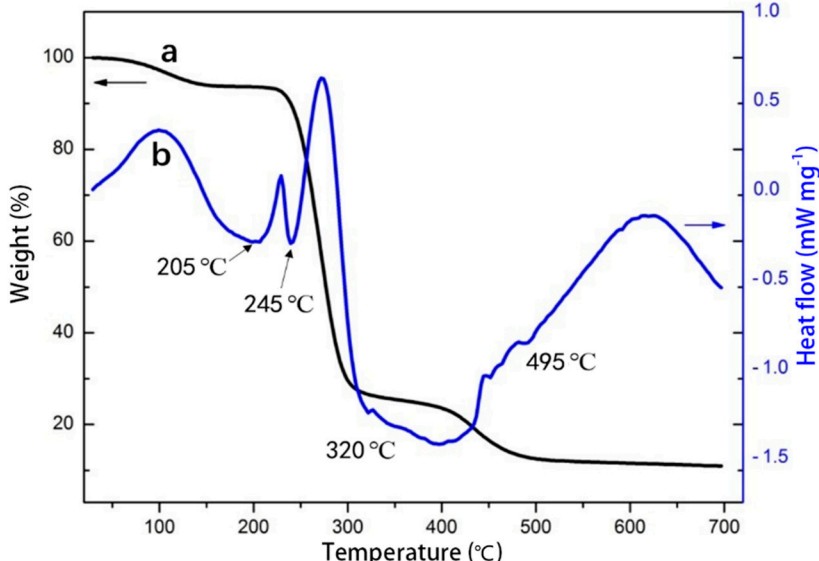

**Figure 3.** TG (**a**) and DSC (**b**) curves of PVA/BC/C-Ag composite.

### 3.2. Mechanical Properties and Swelling Properties of PVA/BC/C-Ag Composite Membranes

Figure 4a shows the stress–strain curve of PVA/C-Ag and PVA/BC/C-Ag composite membranes. The mechanical strength of the composite membranes was significantly improved by the addition of BC, mainly due to the high strength and high modulus of elasticity of the BC nanofibers [26,27]. The mechanical properties of the PVA/BC/C-Ag composite membranes were also improved because BC has many hydroxyl groups, and a large number of hydrogen bonds were formed between BC and PVA.

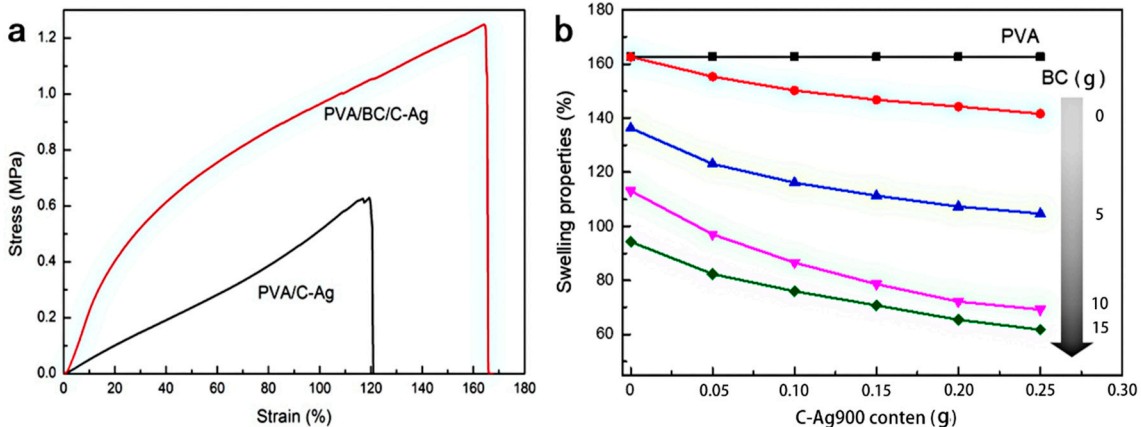

**Figure 4.** Mechanical properties (**a**) and swelling properties (**b**) of PVA/C-Ag and PVA/BC/C-Ag composites and membranes.

The addition of BC and C-Ag reduced the swelling properties of the composite membrane (Figure 4b), likely due to the hydrophobic interactions of C-Ag and to the hydrogen bonding between PVA and BC. The effect of BC content on swelling ratio was greater than the effect of C-Ag. Pure PVA

membranes are easily ruptured after prolonged swelling water and are not suitable for use in drinking water systems. However, the composite membrane with added BC and C-Ag had improved swelling and antibacterial properties that make it applicable to drinking water treatment.

### 3.3. Antibacterial Activity of PVA/BC/C-Ag Composite Membranes

The addition of BC improved the mechanical properties of the composite membrane due to BC's 3D network architecture, and the resulting membrane was denser (Figure 2b). However, the addition of BC did not significantly improve the membranes' antibacterial properties (Figure 5). The addition of C-Ag did improve the membranes' antibacterial properties, and this effect was slightly enhanced as the C-Ag content was increased (Figure 5). The antibacterial zone averaged 16.1 ± 0.3 mm in diameter at a C-Ag content of 0.05 g, and the size of the antibacterial zone increased with C-Ag content. The addition of 0.1 g C-Ag may be most cost-effective, striking a balance between antibacterial performance and cost, as silver is a precious metal. The antibacterial effect of the PVA/BC/C-Ag composite membrane was mainly derived from the silver, as nanosilver ruptures bacterial cell membranes and allows cellular contents to flow out, inhibiting bacterial growth and sterilizing the surrounding medium.

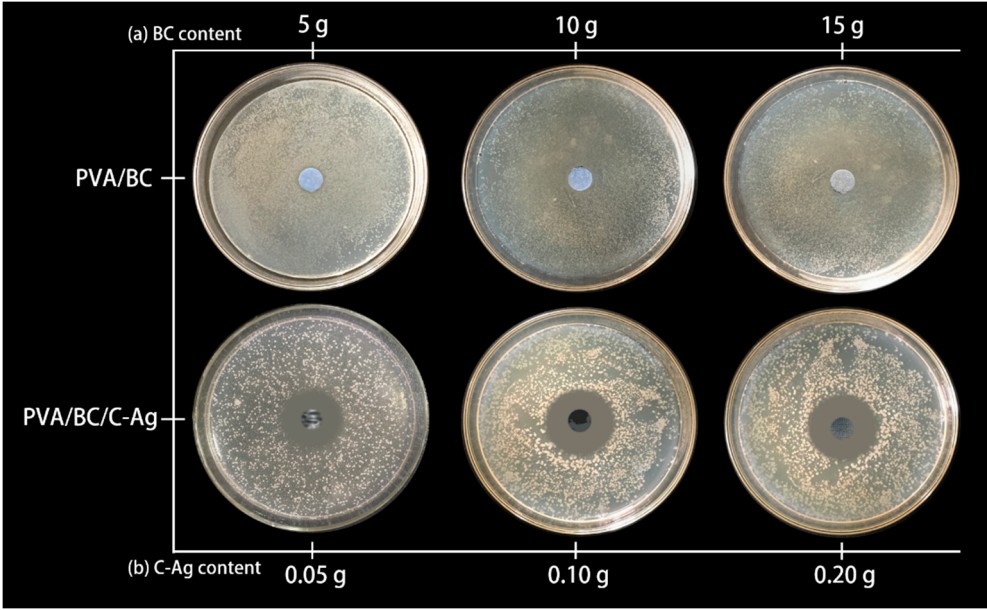

**Figure 5.** Effect of BC (**a**) and C-Ag (**b**) content on antibacterial ability of composite membranes.

### 3.4. Silver Loss from PVA/BC/C-Ag Composite Membranes

The silver bleed curve of the PVA/BC/C-Ag composite membrane is presented in Figure 6. According to the national drinking water standard (GB5749-2006, China), the content of silver in drinking water should not exceed 0.05 mg/L. After 30 days of continuous immersion of the composite membrane, silver concentrations in the water remained lower than the national drinking water standard (Figure 6a). Immersion was repeated 10 times, and the silver loss remained below the standard after each use (Figure 6b). The PVA/BC/C-Ag composite membrane therefore appears highly suitable for practical antibacterial water treatment.

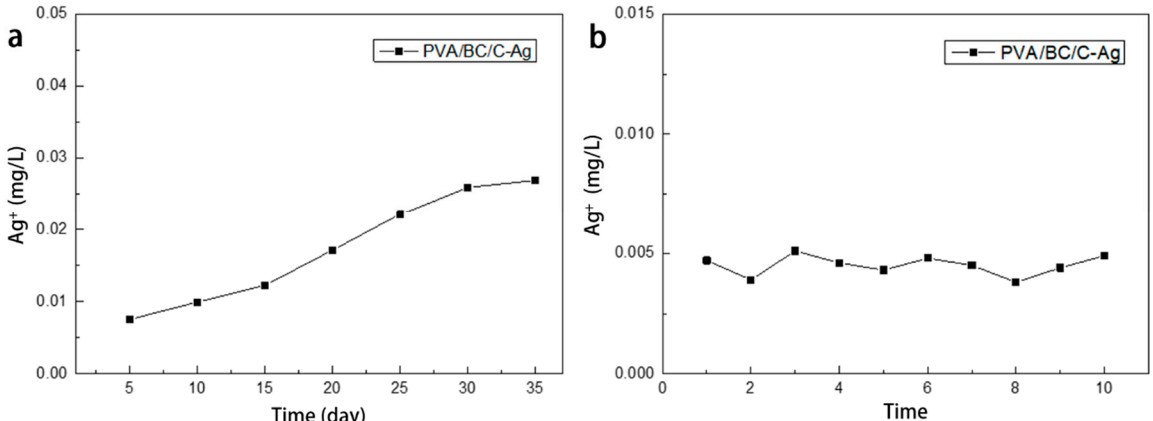

**Figure 6.** Silver loss from PVA/BC/C-Ag composite membranes for continuous (**a**) and repeated use (**b**).

### 3.5. Antibacterial Persistence and Reusability of PVA/BC/C-Ag Composite Membranes

The number of *E. coli* colonies in water after continuous treatment of the composite membrane is shown in Supplementary Table S2. The PVA/BC/C-Ag composite membrane maintained an antibacterial rate of approximately 80% after 30 days of continuous bacteriostasis according to calculations, indicating that it had excellent antibacterial durability (Figure 7a). Some residual bacteria may have adhered to the surface of the composite membrane during the continuous bacteriostatic process, decreasing the bacteriostatic effect. The antibacterial rate of the composite membrane decreased as the number of immersion cycles increased, because the residual deposition of dead bacteria on the surface or internal of the membrane hindered the release of silver ions (Figure 8b). The number of *E. coli* colonies in water after repeated treatment of the composite membrane is shown in Supplementary Table S3. The antibacterial rate was still calculated to be above 85% after repeated use five times, indicating that the composite membrane has satisfactory reusability in practical applications. After each reuse, we measured the concentration of silver in the solution, and in every case the silver concentration was less than 3 µg L$^{-1}$ (Figure 6), below the national drinking water standard (GB5749-2006, China). Silver loss from the PVA/BC/C-Ag composite during the bacteriostatic process was extremely small, and the composite appears to be very safe for drinking water applications.

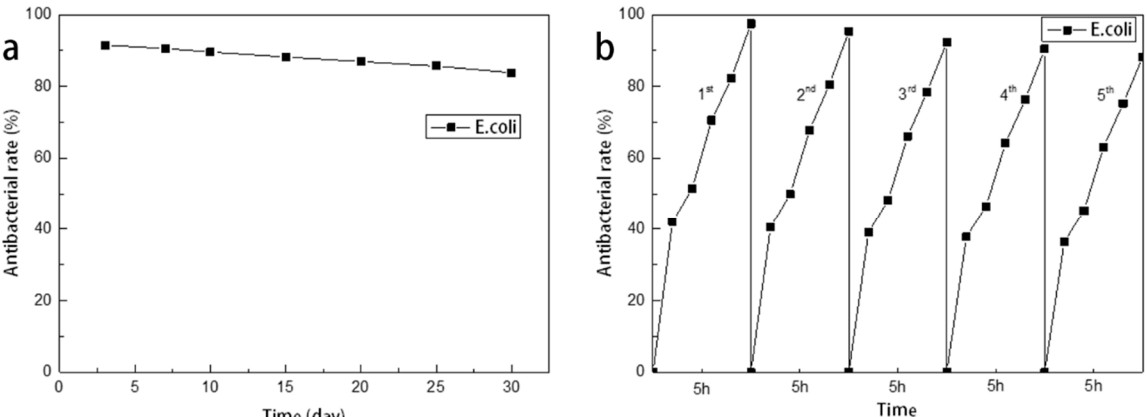

**Figure 7.** Bacteriostatic persistence (**a**) and reusability (**b**) of PVA/BC/C-Ag composite membranes for *Escherichia coli*.

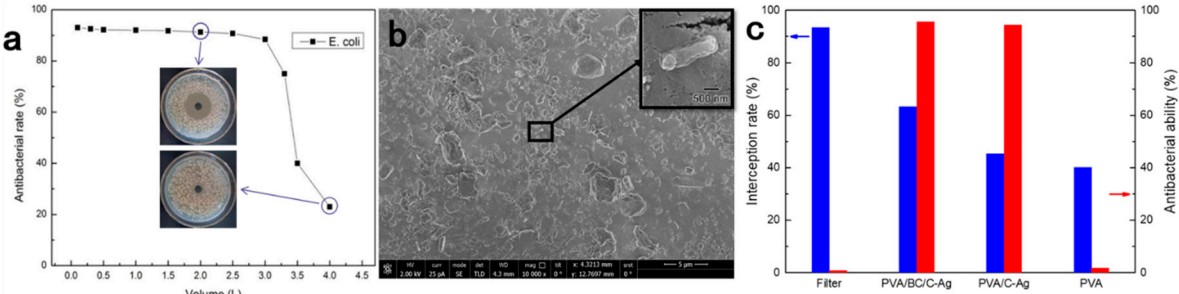

**Figure 8.** PVA/BC/C-Ag composite membrane antibacterial curve for *E. coli* (**a**), SEM of PVA/BC/C-Ag membrane after treatment of contaminated water (**b**), and comparison of the retention rate and antibacterial ability of *E. coli* after filtration of bacterial suspension with commercial filter paper (50 mm in diameter and 0.22 μm in pore size) and several membranes (**c**).

### 3.6. Antibacterial Activity of PVA/BC/C-Ag Composite Membranes in Actual Water

The tap water used was filtered through a 0.45 μm filter (the simulated terminal water had passed the previous strainer and adsorption, and the impurity content is very small). Research showed that the tap water had little effect on the antibacterial performance of the composite membrane (after filtering 5 L). The composite membrane demonstrated excellent antibacterial ability when used in a practical simulation with microbially contaminated water. The bacteriostatic rate remained above 88% when filtering less than 3 L of sewage containing $2.3 \times 10^5$ CFU/mL of *E. coli* (Figure 8a). After filtering 2 L of sewage, a composite membrane was removed and placed on sterilized solid medium coated with *E. coli* and cultured at 37 °C for 24 h. Under these conditions, the PVA/BC/C-Ag composite membrane had a bacteriostatic rate of up to 90%. By comparison, after treating 4 L of sewage, the antibacterial ability of the composite membrane was reduced to approximately 20% (Figure 8a). This decreased bacteriostatic ability may reflect a lower release efficiency of silver ions as the void network became blocked by residual killed bacteria. To assess this hypothesis, a PVA/BC/C-Ag membrane that had filtered bacterial suspension was magnified 10,000 times under an electron microscope (Figure 7b). The morphology of the membrane had changed and unknown substances were visible on the surface; these may have been adsorbed fragments of bacterial cell membranes and cellular contents. The magnified area (Figure 8b inset) showed the remnants of a ruptured *E. coli* bacterium, highlighting the antibacterial mechanism of the composite membrane.

Compared with a water flux of 65,052 Jw/L (m$^2$·h) through the pure PVA membrane, the PVA/BC/C-Ag composite membrane had a flux of 18,039 Jw/L (m$^2$·h), indicating that it had a smaller pore size. However, experiments indicated that the membrane could become blocked due to its water flux drop after 4 L treatment. After filtering 4 L of contaminated water, the composite membrane had lost much of its antibacterial ability and its remaining bacteriostatic properties were mainly due to its retention capacity. PVA quickly swells and is scrapped during use, while composite membranes are sustainable. Indeed, when the membrane was reused, it still retained more than 90% of its antibacterial ability. In actual drinking water, microbial concentrations are generally much lower than in the simulated contamination experiment performed here, and the PVA/BC/C-Ag composite membrane is likely to perform well for extended periods of time in practical applications. Data from the inhibition zone size (Figure 8a) and rejection rate tests (Figure 8c) underscored the composite membrane's excellent antibacterial properties and effective retention performance.

### 3.7. Synergistic and Antibacterial Mechanisms of PVA/BC/C-Ag Composite Membranes

The PVA gel had a 3D network structure that combines with BC, which also had a dense network structure, to form a denser structure (Figure 9). This structure constituted the substrate throughout which silver-loaded biochar was uniformly dispersed to form a PVA/BC/C-Ag composite membrane for bacteriostatic treatment of drinking water. Corn stalks were used as a source of biochar because of

their low cost, environmental friendliness, and sustainability. Because carbonized straw has a large specific surface area, the silver-loaded biochar was prepared for water treatment after loading silver. The main antibacterial effect of the PVA/BC/C-Ag composite membranes derived from the silver, which is widely used due to its broad-spectrum and long-lasting bactericidal properties. The zeta potential analysis showed that the surface of the PVA/BC/C-Ag composite had a positive potential in deionized water (Supplementary Figure S1). The positive charge released by nanosilver destroys bacteria through electrostatic interactions with negative surface charges on cell membranes, causing the cells to rupture. It also interferes with nutrients in the cell, further inhibiting bacterial growth and causing sterilization. A representative ruptured *E. coli* bacterium can be seen in Figure 8b.

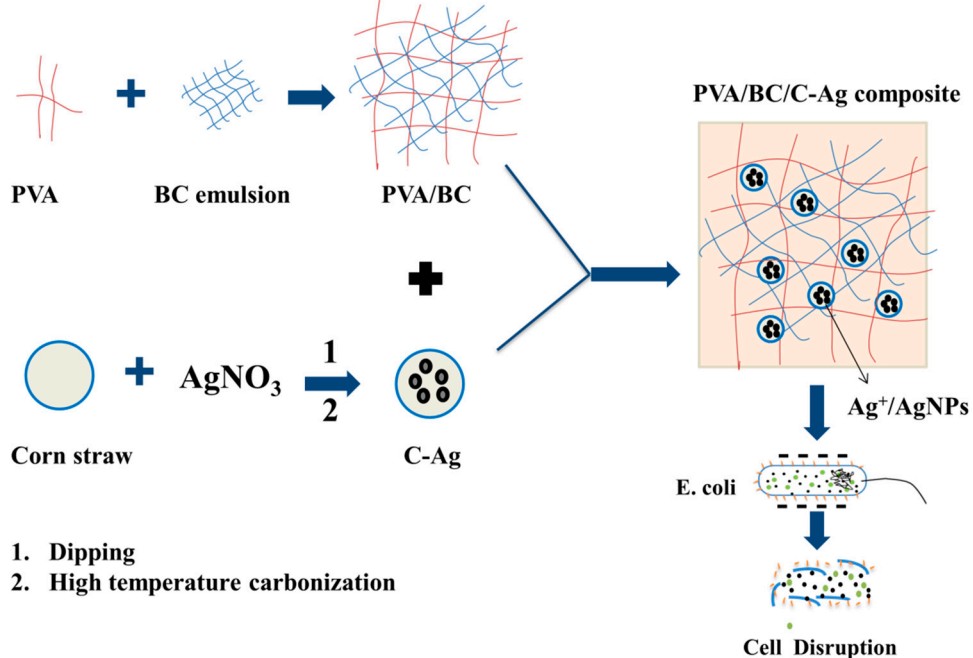

**Figure 9.** Antibacterial mechanisms of PVA/BC/C-Ag composite membrane.

## 4. Conclusions

A PVA/BC/C-Ag composite membrane with excellent antibacterial activity against *E. coli* was synthesized and its three-dimensional network structure was characterized. The addition of BC further strengthened the network structure and improved the mechanical properties of the PVA gel. The composite membranes were analyzed by FT-IR, XRD, SEM, and TG-DSC. The antibacterial properties of Ag were studied and possible bacteriostatic mechanisms proposed. The PVA/BC/C-Ag composite membrane exhibited long-lasting bacteriostatic ability, favorable durability, and resistance to silver loss in a simulated contaminated water application. The composite membrane has excellent prospects for application in drinking water purification systems.

**Supplementary Materials:** The following are available online at http://www.mdpi.com/2076-3417/10/3/752/s1, Table S1: The results of BET measurement, Table S2: Plate count of *E. coli* colonies after continuous treatment, Table S3: Plate count of *E. coli* colonies after repeated treatment, Figure S1: Zeta potential analysis of the PVA/BC/C-Ag composite.

**Author Contributions:** Conceptualization, L.Z., Y.W., X.Z. and J.X.; Project administration, L.Z.; Writing – original draft, Z.H.; Writing—review & editing, S.Z.; Visualization, S.Z.; supervision, L.Z.; project administration, L.Z.; funding acquisition, L.Z. All authors have read and agreed to the published version of the manuscript.

**Funding:** The authors acknowledge partial support from the National Natural Science Foundation of China (51874223, 51874227) and the Natural Science Major Research Plan in Shaanxi Province of China (2017ZDJC-25).

**Conflicts of Interest:** The authors declare no conflict of interest.

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
