# Peer review of "Preparation of Polyvinyl Alcohol/Bacterial-Cellulose-Coated Biochar–Nanosilver Antibacterial Composite Membranes"

_applsci, doi:10.3390/app10030752_

Round 1

Reviewer 1 Report

The work deals with the synthesis and characterization of PVA-BC/Ag membraned that the authors used to remove pathogens. The subject of the work belongs to Applied Sciences (but maybe the work is more appropriate for a more “environmental” journal. However this is an editor decision)

Some points:

The idea is good, and the way that the authors design the experiments, what is missing, in my opinion, is some additional information about the composite. For example, the BET and the zero point charge of the material.

The section of the TG seems like the authors have access to the instrument, and they include the data. Since the process that the authors described is at mild operating conditions, I cannot find why TG is useful here.

I lost somewhere inside the manuscript. From Fig 4 and above the authors choose one specific loading of Ag (and this is not clear to me for the next figures), why and what was the loading?

Figure 8 it is not clear to me why the authors justify the decrease of  the efficiency only due to the “blockage od dead e coli.” There are no other constituents of the WW (like salts, organic matter, solids) to block both the pores or the Ag?

Reviewer 2 Report

Zhang et al investigated the use of polyvinyl alcohol/bacterial cellulose-coated biochar-nanosilver composite membrane for its antibacterial property. The authors also prepared the membrane and characterized it using various instruments.

The study is an important contribution towards finding efficient ways to remove contaminants from drinking water, including pathogenic bacteria.

I would recommend publication of this study, but after the following concerns are addressed and changes are made (or explained):

Section 3.4

 - The silver loss from the composite membrane is a concern. The authors must address the discussion on whether it will exceed the drinking water standard after multiple uses. A recommendation must also be stated to suggest that further research will focus on minimizing the silver loss from the membrane. Some discussion on how the authors plan to minimize silver loss would be worthwhile.

Section 3.5

 - Raw data on cell count (showing the effect of silver) must be shown and discussed.

Figure 7 Replace “bacteriostatic” with “Bacteriostatic” 7(a): y-axis title spelling error Reference Section: Font type and size must be corrected

Author Response

This manuscript is a resubmission of an earlier submission. The following is a list of the peer review reports and author responses from that submission.